# Prediction of strong Cu(I)–He interaction at open metal sites enables isotope-selective helium adsorption

Elvira Gouatieu Dongmo[1,2,3], Shubhajit Das[3], Felix Moncada[3], Toshiki Riemer-Wulf[2] & Thomas Heine [ID][3,4,5] ✉

Helium is generally known as an inert element due to its high ionization potential, zero electron affinity, and low polarizability. Here, we demonstrate that Cu(I) sites with favorably coordinated ligands reach unexpectedly large He binding energies, up to 19 kJ mol$^{-1}$, due to He polarization and charge accumulation along the Cu-He bond. First, we perform accurate electronic structure calculations on a series of Cu(I)-He gas phase clusters to elucidate the nature of the Cu-He interaction. Then, we establish a predictive model to study larger systems hosting Cu(I) sites, including crown ethers, zeolites and metal-organic frameworks (MOFs). The strong Cu(I)-He interaction induces significant differences in the $^4$He/$^3$He zero-point energies, allowing prediction of selective isotope adsorption at technologically relevant temperatures (20–77 K). In particular, undercoordinated Cu(I) sites found in zeolites and MOFs emerge as promising materials with a predicted $^4$He/$^3$He separation factor approaching three at 20 K.

The noble gas helium (He) is known for its chemical inertness. Despite mechanical trapping (caging),[1] chemical interactions of helium seem to be impossible due to its low polarizability of $0.204 \times 10^{-24}$ cm$^3$,[2] about four times smaller than that of H$_2$, its enormously high ionization potential of almost 25 eV, and its zero electron affinity. Therefore, helium is considered to be an inert probe material, used for example to examine the porosity of materials using porosimetry or pycnometry.[3] Trapping helium typically involves tiny cages and gates, where spatial confinement hinders diffusion.[4] Studying the chemical affinity towards helium has been restricted to very few reports,[1] and to date, there are no reports to the best of our knowledge on exploiting chemical affinity to adsorb helium or even to separate its two stable isotopes, which include the most precious isotope on the planet, helium-3 ($^3$He).

$^3$He is a stable isotope with a natural abundance of 1.37 ppm as fraction of the total terrestrial helium occurrence, which is vastly dominated by helium-4, $^4$He.[5–7] Its remarkable properties, in particular its fermionic character, make it a valuable material with significant applications in medicine for cooling MRI scanners, it provides shielding and leak-detection functionality in welding and electronics manufacturing, and enables fundamental studies in cryogenics research.[8–10] Another potential use of $^3$He is as fuel source in deuterium–helium-3 fusion reactors. As these reactors release high-energy protons (compared to neutrons in deuterium-tritium reactors), they can be operated more easily, making this technology an excellent candidate for clean energy generation.[6,11] The increasing global demand for $^3$He,[5,8,12] combined with a lack of sources of production which still mainly rely on tritium decay,[13] call for alternative sources. Separation and purification of $^3$He from natural helium supplies are currently being developed, but still require purification at about 2 K.[14] Prior theoretical studies reported the possibility of $^3$He separation from $^4$He by exploiting kinetic quantum sieving at low temperature ($\leq 20$ K) using nanoporous membranes, specifically nitrogen-functionalized graphene pores.[15–17] Beyond these techniques, differences in zero-point energies (ZPE) between $^3$He and $^4$He can also be utilized through chemical

[1]Wilhelm-Ostwald-Institut für Physikalische und Theoretische Chemie, Universität Leipzig, Leipzig, Germany. [2]Institute of Resource Ecology, Research Site Leipzig, Helmholtz-Zentrum Dresden-Rossendorf, Leipzig, Germany. [3]Faculty of Chemistry and Food Chemistry, School of Science, Technische Universität Dresden, Dresden, Germany. [4]Department of Chemistry and ibs for Nanomedicine, Yonsei University, Seoul, Republic of Korea. [5]Center for Advanced Systems Understanding (CASUS), Görlitz, Germany. ✉e-mail: thomas.heine@tu-dresden.de

affinity quantum sieving (CAQS), which enables isotope separation at significantly higher temperatures than current state-of-the-art approaches.[14] In fact, CAQS has been suggested for dihydrogen isotopologues in various metal-organic frameworks (MOFs)[18-22] and zeolites.[23-25] Among those, materials containing open Cu(I) centers have been found particularly promising due to differences in adsorption enthalpies between $H_2$ and $D_2$ reaching 2.3– 3.0 kJ mol⁻¹ due to nuclear quantum effects (NQEs).[22,26,27]

In this work, we demonstrate surprisingly strong interaction of He with Cu(I) sites, with adsorption energies of up to ~−19 kJ mol⁻¹. The interaction is predominantly governed by electron density sharing as revealed by a topological analysis of the electron density and is sufficiently large to cause reasonable ZPE differences between the isotopes ³He and ⁴He. We first study small model complexes for which it is feasible to apply high-level theory both for the electronic interactions and for the nuclear quantum effects (NQEs), and thus predict the adsorption energies and related isotope separation factors with high precision. We then derive a computational protocol that allows accurate predictions also for larger molecules (crown ethers and a metalloporphyrin) and clusters, the latter ones representing typical Cu(I) adsorption sites in MOFs and in zeolites. The MOF models are obtained by a multistep filtering strategy: by observing the Cu(I) complexes are adsorbing strongest if coordinated to two, three or four ligands, we selecte nine MOF systems created suitable model clusters to study their interaction with He. Among those, Cu(I) hosted on the secondary

building units of a defected MOF, UiO-66, and WOLRIZ are most active, with adsorption energies up to ~−4.0 kJ mol⁻¹, and with a separation factor approaching 3 at LH2 temperature (20 K). These findings are of fundamental interest to chemistry and opens the door for alternative ³He separation techniques.

## Results and Discussion

### Model Cu(I) complexes

We have selected eight Cu(I) complexes for initial conceptual and benchmark studies: two charged clusters Cu⁺He and Cu⁺($H_2O$)He, which would be amenable to assessment in ion trap experiments as demonstrated for dihydrogen isotopes earlier,[28,29] six uncharged ones, Cu⁺X⁻He (X = F, Cl, Br, OH & SH), and Cu⁺($H_2O$)(OH⁻)He (Fig. 1A). All structures have been fully optimized at the CCSD(T)/aug-cc-pVTZ level of theory. In all eight model complexes we observe a strong Cu·He interaction. This is reflected in Born-Oppenheimer (BO) Cu−He bond distances ranging from 1.70 to 2.00 Å and BO adsorption energies (BSSE corrected) of −7.6 to −19.4 kJ mol⁻¹. The strongest interaction is found for the neutral Cu⁺(F⁻) complex, which shows more than twice the adsorption energy compared to the bare Cu⁺ ion (Fig. 1A, Table 1). Note that, Frenking et al. previously reported a similar interaction strength between He and BeO albeit at the MP2 level of theory.[30]

The strong Cu−He bonding interactions are revealed by a quantum theory of atoms in molecules (QTAIM) analysis (PBE0-D3(BJ)/TZ2P, see Methods and Supplementary Table 1), the bond critical

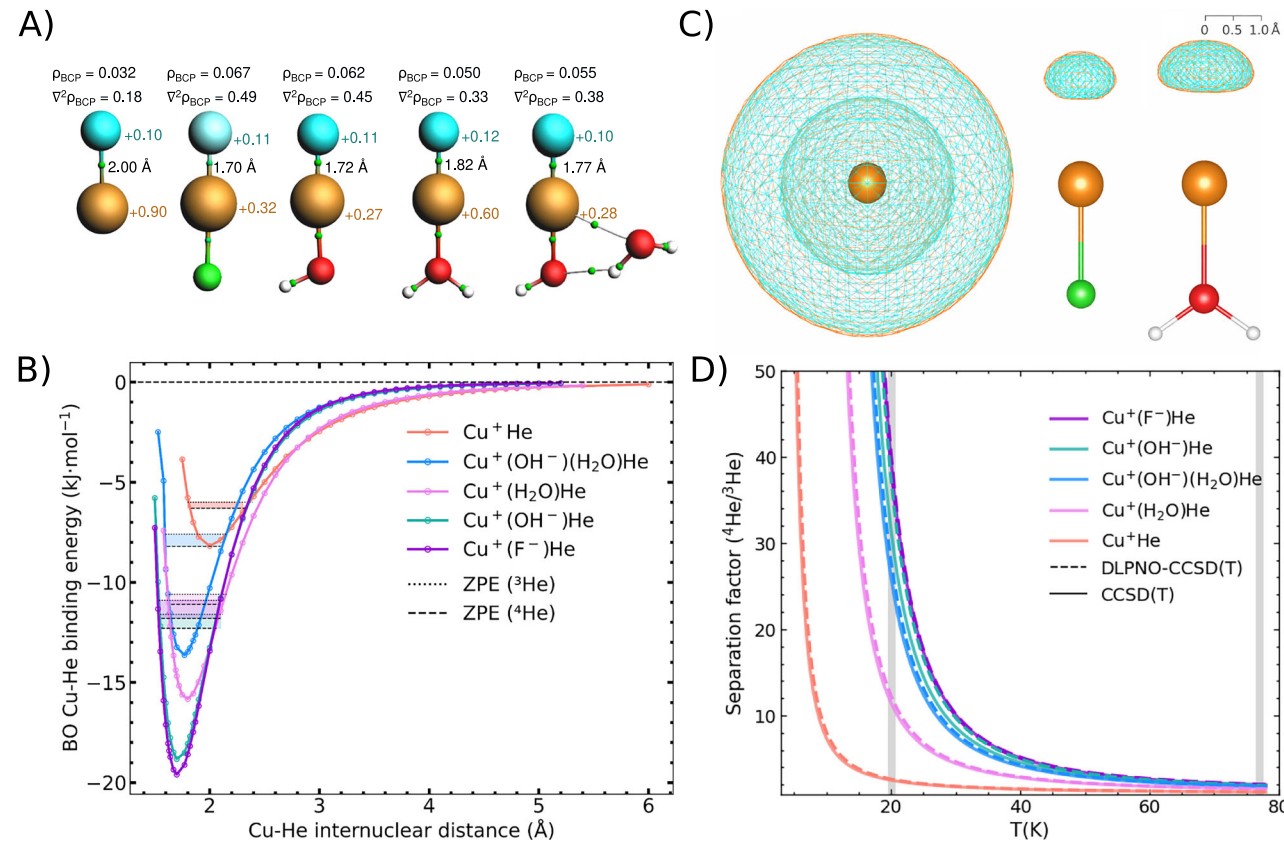

**Fig. 1 | Accurate quantum chemical calculations of Cu−He interactions in Cu(I) complexes. A** Structures of the Cu⁺He, Cu⁺(F⁻), Cu⁺(OH⁻)He, Cu⁺($H_2O$)He and Cu⁺($H_2O$)(OH⁻)He complexes and results of the QTAIM analysis. Cu·He bond lengths (CCSD(T)/aug-cc-pVTZ level) are shown next to the bond paths and critical points as obtained by the QTAIM analysis, while Hirshfeld charges are shown for Cu (in brown) and He (in light blue) (PBE0-D3(BJ)/TZ2P level on CCSD(T)/aug-cc-pVTZ geometries). Colour scheme: Cu-brown, O-red, He-light blue, H-white. Bond critical points are shown as small green spheres. Values of the electron density $\rho_{BCP}$ and its Laplacian $\nabla^2\rho_{BCP}$ are reported in atomic units (a.u.). **B** Plot of the potential energy

along the Cu−He stretch coordinate. All values are at the BSSE-corrected CCSD(T)/aug-cc-pVTZ level. ZPE-corrected adsorption energies (numerical method) are given for ³He (dotted lines) and ⁴He (dashed lines). **C** DVR ⁴He (light blue mesh) and ³He (orange mesh) nuclear density contours of 99% cumulative probability, for Cu⁺He, Cu⁺(F⁻)He and Cu⁺($H_2O$)He. **D** Predicted separation factors for adsorptive He isotopes at low temperatures. All values are based on the CCSD(T)/aug-cc-pVTZ level (solid line) and DLPNO-CCSD(T)/cc-pVTZ(cu·He: aug-cc-pVTZ) dashed line). The grey bars at 20 K and 77 K correspond to liquid hydrogen and liquid nitrogen temperatures, respectively.

**Table 1 | Cu–He interaction analysis**

| Systems | $E_{ads}^{BO}$ | $d_{CuHe}$ | | $E_{ads}^0$ (analytical) | | $E_{ads}^0$ (numerical) | | $q_{CuHe}$ (e) | $q_{He}$ (e) | $\rho_{BCP}$ (e · $r_{Bohr}$) | $\Delta$ZPE | |
|---|---|---|---|---|---|---|---|---|---|---|---|---|
| | | $^3$He | $^4$He | $^3$He | $^4$He | $^3$He | $^4$He | | | | analytical | numerical |
| Cu$^+$He | −7.6 [−7.2] | 2.08 [2.07] | 2.07 [2.06] | −6.0 [–] | −6.2 [–] | −6.1 [−5.7] | −6.3 [−5.9] | +0.900 [+0.897] | +0.101 [+0.103] | 0.032 [0.034] | 0.20 [–] | 0.18 [0.19] |
| Cu$^+$(F$^-$)He | −19.4 [−18.6] | 1.74 [1.73] | 1.73 [1.72] | −13.8 [–] | −14.6 [–] | −14.1 [−13.1] | −14.8 [−13.8] | +0.319 [+0.332] | +0.112 [+0.096] | 0.067 [0.036] | 0.74 [–] | 0.68 [0.68] |
| Cu$^+$(OH$^-$)He | −17.1 [−16.0] | 1.77 [1.76] | 1.76 [1.75] | −11.7 [–] | −12.4 [–] | −11.9 [−10.5] | −12.6 [−11.2] | +0.265 [+0.264] | +0.106 [+0.108] | 0.062 [0.066] | 0.71 [–] | 0.64 [0.67] |
| Cu$^+$(H$_2$O)He | −14.4 [−13.1] | 1.86 [1.85] | 1.85 [1.84] | −10.7 [–] | −11.2 [–] | −10.5 [−9.1] | −11.0 [−9.6] | +0.599 [+0.597] | +0.120 [+0.122] | 0.050 [0.052] | 0.49 [–] | 0.48 [0.49] |
| Cu$^+$(H$_2$O)(OH)He | −12.1 [−11.0] | 1.82 [1.80] | 1.81 [1.79] | −7.2 [–] | −8.2 [–] | −7.2 [−6.2] | −7.7 [−6.8] | +0.284 [+0.278] | +0.100 [+0.104] | 0.055 [0.059] | 0.59 [–] | 0.61 [0.62] |

Adsorption energies $E_{ads}$, and Cu-He bond lengths ($d_{CuHe}$) at the BSSE-corrected CCSD(T)/aug-cc-pVTZ level of theory (values given in the square brackets). Nuclear quantum effects are accounted for using the numerical approach (see text). For comparison, ZPE corrections using the harmonic approximation are given for the CCSD(T) calculations. Hirshfeld charges for Cu in the systems with adsorbed helium $q_{CuHe}$ and for the adsorbed He atom $q_{He}$, electron density $\rho_{BCP}$ at the BCP between Cu$^+$ and He. The ZPE difference is also benchmarked for calculating Cu$^+$-He adsorption energies. All distances are expressed in Å and energy values in kJ mol$^{-1}$.

points are located halfway between Cu and He. Electron density and Laplacian of electron density values for the Cu$^+$(F$^-$)He system being more than twice and five times as large as for the hydrogen bond in a water dimer. Compared to the nonpolar Ar dimer, these values are significantly higher by a factor of 20 or more (Fig. 1A, see also Supplementary Fig. 3). This indicates an usually strong polarization-driven interaction; as well as a slight covalent bonding nature.[31] Moreover, charge transfer of about 0.1 electrons from the He to Cu centers (Hirshfeld charges, see Table 1) cause electrostatic attraction. Indeed, the most strongly bound complexes Cu$^+$(F$^-$)He and Cu$^+$(OH$^-$)He show the largest density and Laplacian values at the bond critical points. In addition, energy decomposition analyses (EDA-NOCV)[32,33] reveal that, within the neutral complexes bearing anionic ligands, the binding energy follows the electronegativity order of the donor atom bonded to Cu: lower electronegativity leads to weaker binding, consistent with the decreasing electrostatic contribution to the total interaction energy (see Supplementary Table 2).

To assess realistic materials models, the CCSD(T)/aug-cc-pVTZ level of theory is prohibitively expensive. We replace it with the computationally substantially more efficient DLPNO-CCSD(T) method,[34,35] and use for all but the bond-related Cu and He atoms the smaller cc-pVTZ basis set. The results are in close agreement with the parent method, with Cu-He bonds differing by 0.02 Å or less, and adsorption energies by 1.3 kJ mol$^{-1}$ or less (Table 1). Thus, DLPNO-CCSD(T) offers a computationally efficient yet reference-grade protocol for larger cluster models with under-coordinated Cu(I) sites.

To analyze the NQEs on He adsorption, vibrational frequencies have first been assessed analytically using the harmonic approximation (Supplementary Table 3). However, the potential energy surface (PES) of the Cu−He coordinate has a strongly anharmonic character (Fig. 1B), leading to asymmetric distributions of the He nuclear densities along the Cu-He axis (Fig. 1C). Moreover, the Cu-He bond dissociation curve expectedly shows the strongly anharmonic character of weak bonds, with the different asymptotic character of the charge neutral (London dispersion governed ~ r$^{-6}$ asymptotics) compared to the charged (charge-induced dipole governed ~r$^{-4}$ asymptotics) clusters. As standard methods to account for anharmonicity are computationally unfeasible for the realistic material models, we developed an approach focused on the most significant degrees of freedom for He adsorption: the Cu−He bond stretch and the two He displacements normal to this bond. For the Cu-He stretch NQEs, first we fitted the 1D PES along this coordinate to a Morse potential augmented with a charge-induced dipole term, and then solve the 1D Schrödinger equation numerically using the finite difference method (FDM).[36,37] The NQEs of the two He degrees of freedom normal to the Cu-He bond were described using a quartic anharmonic correction to a harmonic oscillator potential. We call the combination of the FDM plus quartic NQE models the "numerical" approach in the remainder of the document. The results obtained with this numerical approach have been validated against second order vibrational perturbation theory (VPT2) and discrete variable representation (DVR) calculations (for details see Supplementary Notes 2 and 5)

NQEs of all complexes have been calculated for the two stable isotopes of helium, $^3$He and $^4$He (Table 1). Quantum effects reduce the adsorption energies by 1.3 kJ mol$^{-1}$ for Cu$^+$-$^4$He and 4.5 kJ mol$^{-1}$ for Cu$^+$(OH$^-$)-$^4$He, and thereby slightly increase Cu-He distances (see Table 1). The ZPE increases, as expected, with the adsorption energy, but is also affected by the long-range asymptotics of the PES which shows a steeper decline for the charge-neutral complexes. This is reflected in the ZPE difference ($\Delta$ZPE) between the $^3$He and $^4$He isotopes, which strongly increases from 0.48 kJ mol$^{-1}$ for Cu$^+$(H$_2$O)He to 0.64 kJ mol$^{-1}$ for Cu$^+$(OH$^-$)He. The pronounced $\Delta$ZPE result in $^4$He/$^3$He separation factors reaching 39.5 at liquid hydrogen (LH2) and even 2.0 at liquid nitrogen (LN2) temperatures for Cu$^+$(F$^-$)He (Fig. 1C and

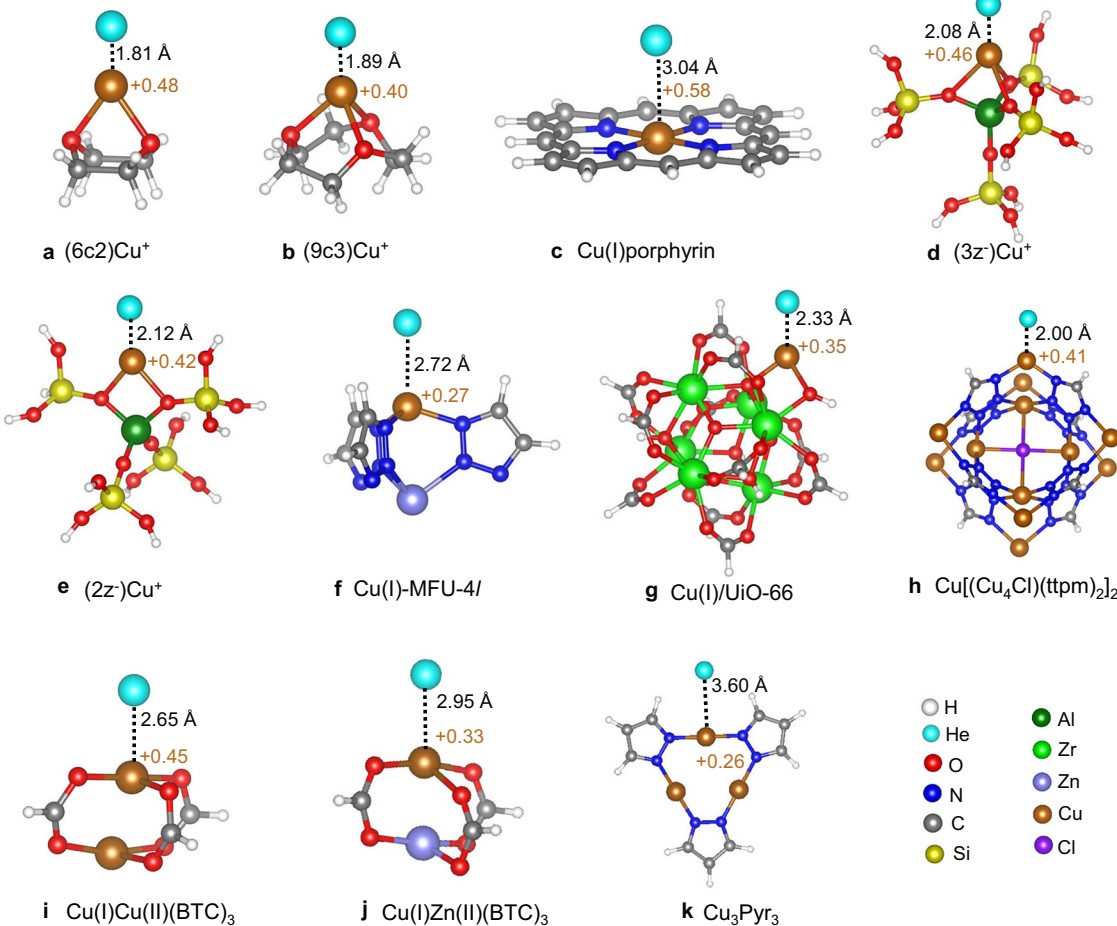

**Fig. 2 | Atomistic materials models with adsorbed He atom at Cu(I) sites, indicating Cu–He bond lengths (black) and copper partial charges (brown).** Crown-ether models: (**a**) 6-crown-2 (6c2), (**b**) 9-crown-3 (9c3). (**c**) Cu(I)porphyrin. 5 T zeolite cluster models with Cu$^+$coordinated to: (**d**), three oxygen atoms [(3z$^-$) Cu$^+$] and (**e**), two oxygen atoms [(2z$^-$)Cu$^+$]. Small structural models used to represent the SBUs of: (**f**), Cu(I)-MFU-4l; (**g**), Cu(I)/UiO-66; (**h**), Cu[(Cu$_4$Cl)(ttpm)$_2$]$_2$ (ttpm = Tetrakis(4-tetrazolylphenyl)methane); (**i**), Cu(I)Cu(II)(BTC)$_3$; (**j**), Cu(I)Zu(II)(BTC)$_3$ with Zn substitution and (**k**), Cu$_3$Pyr$_3$ (pyr = pyrazolate).

Supplementary Table 5). We note that the adsorption is strong enough that the nearly all Cu(I) sites are occupied at these temperatures.

**Materials Models**

Motivated by the strong Cu(I)–He interactions in the small gas phase model complexes we proceed to investigate the Cu(I)–He interaction in various cluster models of molecules and materials, i.e., crown ethers, porphyrins, zeolites, and metal-organic frameworks (MOFs). For all forthcoming calculations, we are using the established protocol: DLPNO-CCSD(T)/cc-pVTZ (aug-cc-pVTZ for Cu, He) level of theory, with the numerical method to account for NQEs.

The small, computationally feasible undercoordinated Cu(I) complexes of crown ethers, 6-crown-2 (6c2) and 9-crown-3 (9c3) (Fig. 2a, b), have been studied earlier for their hydrogen isotopologue adsorption.[38] In both cases, helium binds strongly to Cu(I). The Cu–He bond lengths are 1.81 Å and 1.89 Å for the (6c2)Cu$^+$ and (9c3)Cu$^+$ complexes, respectively, which is similar as for the small model Cu(I)-complexes discussed above. The BO adsorption energies are −12.4 and −8.4 kJ mol$^{-1}$ (Supplementary Table 10) and weaken to −9.5 (−9.1) and −6.0 (−5.7) for $^4$He ($^3$He) when the ZPE is taken into account (Table 2). QTAIM analysis shows (Supplementary Table 7) slightly weaker charge redistribution and density accumulation in the Cu-He bond which are consistent with these values. Consequently, the differences in ZPE, ΔZPE, are 0.42 and 0.33 kJ mol$^{-1}$ for the two clusters, resulting in remarkable $^4$He/$^3$He separation factors of 8.0 (1.4) and 4.7 (1.2) at LH2

(LN2) temperature (Supplementary Table 10). The calculations suggest that both doubly and triply coordinated Cu(I) sites attract He strongly, with preference for the doubly coordinated site.

A tetracoordinated, planar Cu(I) can be realized, for example, in a metalloporphyrin (Fig. 2c). Metalloporphyrins have been used as linkers in various MOFs, in particular in surface-mounted MOFs where indium oxide rods are cross-linked with tetracoordinated metalloporphyrin linkers, so that the metal sites face a sufficiently large pore volume.[39] However, Cu(I)porphyrin shows very little attraction towards He (Table 2), with a Cu-He distance greater than 3.0 Å. We account this result to the planarity of Cu(I), and will not consider further tetracoordinated Cu(I) configurations.

Based on an earlier work[38] we consider two zeolite model clusters with five tetrahedral sites (5 T) that host undercoordinated Cu(I) centers (Fig. 2d-e). In the first one, Cu(I) is coordinated by two oxygen atoms (2z$^-$). This model is representative for common zeolites such as CHA, LTA und RHO, whereas the tri-coordinated model (3z$^-$) is representative for, e.g., zeolite LTA. Both zeolite models show weaker attraction to helium compared to the crown ethers. The Cu–He bond length exceeds 2 Å (2.12 for the 2z$^-$ and 2.08 for the 3z$^-$ model), resulting in BO adsorption energies of −3.3 and −4.7 kJ mol$^{-1}$ for the two complexes (Supplementary Table 10), respectively. NQEs further reduce the adsorption energy to −2.5 (−2.4) for the 2z$^-$ model and −3.3 (−3.1) for the 3z$^-$ model for $^4$He ($^3$He). This results in ΔZPE of 0.20 and 0.11 kJ·mol$^{-1}$, which results in

**Table 2 | Geometrical parameters of He interacting with models of host materials**

| Systems | $d_{CuHe}$(Å) | | $E_{ads}^0$ (kJ·mol$^{-1}$) | | $q_{CuHe}$(e) [$q_{He}$(e)] | $\rho_{BCP}$ (e · $r_{Bohr}$) | $\nabla^2_{BCP}$ (e · $r_{Bohr}$) |
|---|---|---|---|---|---|---|---|
| | $^3$He | $^4$He | $^3$He | $^4$He | | | |
| **6c2** | 1.90 | 1.89 | −9.1 | −9.5 | +0.481 [+0.110] | 0.052 | 0.334 |
| **9c3** | 1.97 | 1.96 | −5.7 | −6.0 | +0.395 [+0.093] | 0.042 | 0.261 |
| **2z⁻** | 2.23 | 2.21 | −2.4 | −2.5 | +0.417 [+0.067] | 0.023 | 0.130 |
| **3z⁻** | 2.20 | 2.18 | −3.1 | −3.3 | +0.458 [+0.072] | 0.025 | 0.146 |
| **Cu(I)-MFU−4*l*** | 2.94 | 2.92 | −1.0 | −1.1 | +0.267 [+0.019] | 0.006 | 0.026 |
| **Cu(I)Porphyrin** | 3.17 | 3.15 | −2.1 | −2.1 | +0.583 [+0.023] | 0.003 | 0.012 |
| **Cu(I)Cu(II)(BTC)₃** | 2.76 | 2.74 | −1.3 | −1.5 | +0.451 [+0.034] | 0.007 | 0.030 |
| **Cu(I)Zn(II)(BTC)₃** | 3.12 | 3.10 | −0.4 | −0.5 | +0.327 [+0.022] | 0.004 | 0.014 |
| **Cu(I)/UiO-66** | 2.35 | 2.33 | −1.9 | −2.1 | +0.354 [+0.053] | 0.014 | 0.071 |
| **Cu[(Cu₄Cl)(ttpm)₂]₂** | 2.11 | 2.09 | −3.9 | −4.1 | +0.406 [+0.085] | 0.032 | 0.186 |

Bond lengths and energies at the DLPNO-CCSD(T)/cc-pVTZ (aug-cc-pVTZ for Cu, He) level of theory. For Cu(I)/UiO-66•He an effective core potential (ECP) was used for Zr atom. QTAIM properties calculated at the PBE0-D3(BJ)/TZ2P level; Hirshfeld Charges of the cluster models are given with adsorbed He $q_{CuHe}$ (the charges of adsorbed He atom $q_{He}$ are given in the square brackets). Electron Density $\rho_{BCP}$ and its Laplacian $\nabla^2_{BCP}$ at the BCP between Cu$^+$ and He.

$^4$He/$^3$He separation factors of 2.2 and 1.5 for 3z⁻ and 2z⁻, respectively, at LH2 temperature.

MOFs offer an unprecedented diversity of structures, some of which can host Cu(I) sites. To assess these extended framework structures, suitable cluster models have been chosen for each MOF structure discussed below, which are described in detail in the Methods section. Among the MOFs, Cu(I)-MFU-4*l* has been studied intensively for its strong adsorption towards H₂[27,40] and its resulting separation capability of dihydrogen isotopologues.[26] Contrary to the strong interaction to H₂, there is only very weak attraction towards helium as indicated by a Cu-He distance of 2.72 Å, and a BO adsorption energy of −1.8 kJ·mol$^{-1}$. The resulting small ZPE differences of 0.10 kJ·mol$^{-1}$ result in a separation factor of 1.3 at LH2 temperature. Another MOF well-investigated for its catalytic activity is Cu(I)/UiO-66[41] which can anchor isolated Cu(I) sites on defected Zr₆ nodes. These sites have been reported to be catalytically active for reactions such as CO oxidation.[42] This site binds helium more strongly compared to Cu(I)-MFU-4*l*, with a Cu−He distance of 2.33 Å and the corresponding BO adsorption energy of −3.4 kJ·mol$^{-1}$. The corresponding ZPE difference amounts to 0.20 kJ·mol$^{-1}$, which is comparable with that seen in zeolite 3z⁻, resulting in a similar expected performance for $^4$He/$^3$He separation. Furthermore, vibrational mode decomposition analysis confirms that the Cu-He stretch is highly localized, with negligible coupling to the node or lattice vibrations, indicating that the Cu-He interaction is not expected to be significantly modulated by framework dynamics. (see Supplementary Fig. 7). Note that, the same conclusions have been drawn earlier for the adsorption of stronger interacting dihydrogen in various MOFs, e.g. in Cu(I)-MFU-4*l*.[26]

To take advantage of the rich structural diversity of MOFs, we identified additional undercoordinated open Cu(I) sites by screening 952 inorganic secondary building units (SBUs) from the publicly available HEALED library,[43] which comprises a comprehensive set of experimentally synthesized MOF building blocks. Our initial filter targeted SBUs containing Cu in the +1 oxidation state, yielding 31 candidates. Each candidate was then subjected to a coordination geometry analysis to determine which Cu(I) centers constitute plausible open metal sites. Sites with two or three coordinating ligands are classified

as open by default. For four-coordinate complexes, we only considered them as open if their geometries adopt a seesaw or square planar arrangement. After an inspection of the structures identified, we select five SBUs – two bicoordinate, one tricoordinate and two tetracoordinate Cu(I) sites. We refer to these SBUs according to their identifiers in the HEALED library. Additionally, we consider two Cu(I) containing SBUs from the defected paddlewheel nodes of the Cu-BTC MOF, as they have been found to be highly active towards CO adsorption.[44] All these SBUs except one show weak to no attraction towards helium as indicated by the Cu−He bond distance. Interestingly, Cu[(Cu₄Cl)(ttpm)₂]₂ with [ttpm = Tetrakis(4-tetrazolylphenyl) methane][45] (correspond CCDC entry WOLRIZ) shows an especially strong attraction for He as indicated by a Cu-He distance of 2.0 Å and BO adsorption energy of −5.9 kJ·mol$^{-1}$ which is highest among all the zeolites and MOFs considered. The active site in Cu[(Cu₄Cl)(ttpm)₂]₂ is a bicoordinated Cu(I) bound to two tetrazolate ligands. The high activity of Cu[(Cu₄Cl)(ttpm)₂]₂ is attributed to the bent geometry around the active site with a N-Cu-N angle of 96.8° which facilitates He adsorption. By comparison, Cu₃Pyr₃, which also features a very similar chemical environment - bicoordinated Cu(I) connected to pyrazolate ligands - adopts a linear geometry with N-Cu-N angle of 179.7° and as a result barely binds to He (BO adsorption energy 0.8 kJ mol$^{-1}$, Supplementary Table 10). The ZPE difference between the two helium isotopes in Cu[(Cu₄Cl)(ttpm)₂]₂ is 0.24 kJ·mol$^{-1}$ yielding separation factors of 2.8 (1.1) at LH2 (LN2) temperature (Fig. 3 and Supplementary Table 11).

We have shown that helium, which is usually considered to be chemically inert, can have sizable interactions with Cu(I) sites. These polarization-based interactions are enforced by charge transfer from He to Cu. To illustrate the strength of interaction, the electron density given by a QTAIM analysis yields a value greater than twice the hydrogen bond in the water dimer and over twenty times that in the argon dimer. Complexes with Cu−He bond lengths of ~2 Å or less can show appreciable Born-Oppenheimer adsorption energies, reaching −5.9 kJ mol$^{-1}$ for the best materials model, MOFs based on the SBU found in WOLRIZ. However, this value is three times lower than that reported for small gas-phase complexes containing Cu(I) − with a

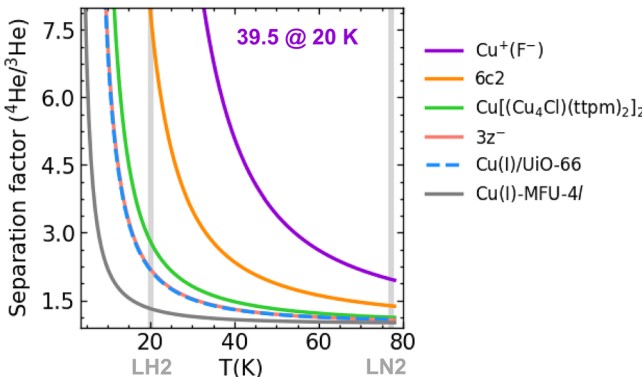

**Fig. 3 | Separation factors of top-performing Cu(I)-hosting materials for He isotope separation.** Predicted separation factors for adsorptive He isotope on selected Cu(I)-candidates, plotted as a function of temperature. The grey bars at 20 K and 77 K correspond to liquid hydrogen and liquid nitrogen temperatures, respectively.

record high of −19.4 kJ mol⁻¹ for Cu⁺(F⁻). This comparison indicates that further exploration may uncover MOFs featuring active, under-coordinated metal sites that are even more promising for adsorptive He isotope separation. Notably, our results highlight that two-coordinated open metal sites in bent geometries show the highest potentials among the MOF models investigated in this work.

The strong interaction energy gives rise to large nuclear quantum effects, sufficient to separate helium isotopes at the technically meaningful temperature of 20 K, the boiling temperature of $H_2$. The separation factor of 2.8 estimated for the best-performing MOF significantly exceeds the state of the art, which requires temperatures below 4 K and shows separation factors of 2 or less. We note that at the relevant temperatures for helium isotope separation, typical obstacles of highly reactive MOFs, namely the adsorption of residual water or other molecules or mechanical stability, are irrelevant. The same is true for the materials costs, as ³He is one of the most expensive isotopes with increasing demand meeting a reduction of traditional production via the decay of tritium obtained from radioactive waste.

To go beyond the Langmuir model employed in this work, further studies require multi-particle effects, including He tunneling between metal sites and fermion (³He)/boson (⁴He) spin effects on adsorption. We note that while this work concentrates on Cu(I), it is likely possible that other active metal centers will attract helium with similar or even superior strength. We hope that this work will start a research endeavor with the aim to do chemistry with the most chemically innocent element on the planet, helium.

## Methods

### Gas phase model clusters
All Cu(I) complexes were fully optimized at the coupled-cluster singles, doubles, and perturbative triples excitations, CCSD(T), level of theory in conjunction with the augmented correlation-consistent polarized valence triple-ζ (aug-cc-pVTZ) basis set. Harmonic vibrational frequency calculations were then carried out for the fully optimized structures with and without adsorbed helium. ZPE were calculated (reported solely in Supplementary Table 4b) from the harmonic analysis. For energy calculations, including single-point calculations to establish the Cu-He bond potential, we corrected for basis set superposition errors in between He and the Cu(I)-hosting fragment. Calculations were performed with the Gaussian 16 program package.[46]

The analysis of the electron density was carried out using QTAIM approach. Calculations using the hybrid PBE0-D3(BJ) level with the Slater-type TZ2P basis set were performed with the Amsterdam Density Functional (ADF 2025) program.[47,48]

DLPNO-CCSD(T) single-point calculations were carried out in ORCA 5.0.1,[49] using both TightSCF/TightPNO threshold settings (i.e, TightPNO settings with $T_{cutMKN}$, $T_{cutPNO}$ and $T_{cutPairs}$ tightened to 10⁻⁶, 10⁻⁸, 10⁻⁷ respectively).[50,51] For the detailed explanation of these cutoff parameters, see refs. [52,53]. While we describe Cu and He with the aug-cc-pVTZ basis as in the canonical CCSD(T) calculations, we renounce the augmentation and use the cc-PVTZ basis for the other atoms for computational efficiency.

### Nuclear quantum effects and isotope separation factors
For the NQE calculations accounting for anharmonicity, we only consider three degrees of freedom (DOF), namely those related to the helium motion. For the stretch vibration, we first fitted the PES of the Cu-He bond, from 1.4 Å to bond dissociation, to a one-dimensional Morse potential with a charge-induced dipole (CID) term, which is switched off for short distances via a Fermi function,

$$V_{MCID}(z) = V_{Morse}(z) + (1 - F_{fermi}(z)) V_{CID}(z) \qquad (1)$$

See Supplementary Note 1 for the definition of each term and Supplementary Tables 14, 16 and 18 for the fitted parameters.

The finite difference method (FDM)[36,37] was then applied to solve the one-dimensional Schrödinger equation using the reduced mass of the Cu-He moiety, with uniform grid spacing and a stepsize of 0.001Å (see details in the Supplementary Note 2). The increase of Cu-He bond length is an additional result of the FDM calculation.

For the two He normal modes to the Cu-He bond, the potential was fitted to a quartic polynomial, via a five-point fit. From this fit, a quartic anharmonicity correction term to the harmonic ZPE was derived. For the ground state, the quartic energy correction is,

$$\Delta E_0 = \frac{3\gamma \hbar^2}{4\mu k} \qquad (2)$$

where $\mu$ is the reduced mass, $k$ is the effective spring constant, and $\gamma$ the quartic correction constant. See additional details in Supplementary Note 2 and the fitted parameters in Supplementary Tables 15, 17 and 19.

The protocol to calculate the numerical frequencies has been validated with second-order vibrational perturbation theory (VPT2) and discrete variable representation (DVR) calculations. The VPT2 analysis was performed at the MP2/def2-TZVPP level to assess the magnitude of the anharmonicity in the vibrational frequencies and the impact of considering only the three DOF related to the helium motion. The MP2 calculations have been performed using Gaussian 16.[46] The DVR calculations, which numerically solve the 3D Schrödinger equation, were performed in cubic grids of CCSD(T)/aug-cc-pVTZ energies with 0.1 Å step size. The DVR results were employed to validate the separation of the Cu-He stretch and the two He normal modes. The NuSol v. 1.0 package[54] was employed for the DVR calculations. Table 3 presents a comparison of the vibrational frequencies for four systems. The complete validation analysis is found in the Supplementary Note 5.

The ⁴He/³He separation factors (⁴He/³He) are calculated via the equilibrium adsorption constants, $K_{ad}(^4He)$ and $K_{ad}(^3He)$. These adsorption constants are obtained from the partition functions, which have been calculated by multiplying the individual contributions from each of the three He normal modes based on the Langmuir model and the ideal gas approximation for translational partition function of free He,

$$\alpha(^4He/^3He) = \frac{K_{ad}(^4He)}{K_{ad}(^3He)} = \frac{q_{str,\,^4He}^{ad}\,q_{\perp_1,\,^4He}^{ad}\,q_{\perp_2,\,^4He}^{ad}}{q_{str,\,^3He}^{ad}\,q_{\perp_1,\,^3He}^{ad}\,q_{\perp_2,\,^3He}^{ad}} \left( \frac{M_{^3He}}{M_{^4He}} \right)^{3/2} \qquad (3)$$

**Table 3 | Calculated vibrational frequencies in Cu(I) complexes**

| Level | | MP2 def2-TZVPP | | | | | CCSD(T) aug-cc-pVTZ | |
|---|---|---|---|---|---|---|---|---|
| Systems | Modes | $\nu_{harm}^{full}$ | $\nu_{harm}^{He}$ | $\nu_{VPT2}^{full}$ | $\nu_{VPT2}^{He}$ | $\nu_{FDMq}^{He}$ | $\nu_{FDMq}^{He}$ | $\nu_{DVR}^{He}$ [b] |
| $Cu^+He$ | $v_s$ | 187.4 | 187.4 | 134.1 | 134.1 | 140.1 | 110.7 | 111.0 |
| $Cu^+(F^-)He$ | $v_s$ | 518.9 | 527.8 | 427.9 | 431.9 | 495.6 | 430.9 | 401.1 |
| | $v_{\perp 1}$ | 215.7 | 180.1 | 199.1 | 168.8 | 185.8 | 159.0 | 150.5 |
| | $v_{\perp 2}$ | 215.7 | 180.1 | 199.1 | 168.8 | 185.8 | 159.0 | 150.5 |
| $Cu^+(H_2O)He$ | $v_s$ | 337.9 | 359.4 | 272.3 | 277.2 | 301.2 | 324.8 | 316.9 |
| | $v_{\perp 1}$ | 118.7 | 101.0 | 109.0 | 89.5 | 100.4 | 100.7 | 84.7 |
| | $v_{\perp 2}$ | 122.4 | 103.8 | 138.3 | 92.5 | 107.0 | 101.2 | 87.9 |

Harmonic, anharmonic VPT2 and FDM plus quartic numerical frequencies (cm$^{-1}$) of $^4$He in small Cu(I)-complexes computed at MP2/def2-TZVPP level of theory. FDM plus quartic numerical and 3D-DVR frequencies at the CCSD(T)/ aug-cc-pVTZ level of theory are also given[a].

[a] $\nu^{full}$ frequencies obtained with full normal modes, $\nu^{He}$ keeps only the three degrees of freedom of the adsorbed He atom, $\nu_{FDMq}^{He}$ are the frequencies obtained using the FDM plus quartic numerical approach. $v_s$ represents the Cu-He stretching vibrational frequency, and $v_{\perp 1}$ and $v_{\perp 2}$ correspond to the orthogonal contribution modes of He atom.

[b] 3D-DVR frequencies are obtained as energy differences between the He vibrational ground state and excited states displaying a nodal plane perpendicular ($v_s$) or parallel ($v_\perp$) to the Cu-X axis.

With

$$q_{m,i}^{ad} = \frac{\exp(-E_{m,i}(0)/RT)}{1 - \exp(-2 \cdot E_{m,i}(0)/RT)}, \tag{4}$$

where R is the ideal gas constant, $T$ is the temperature, $E_{m,i}(0)$ is the ground state energy of the mode $m$ and isotope $i$ and $M$ is the molar mass of the He isotope. For additional details see Supplementary Note 3.

## Material models

The crown ethers, Cu(I)-porphyrin, zeolites and Cu(I)-MFU-4l[27] clusters were all taken from the literature.[38,55] He atom was initially positioned above the Cu(I) site at a reasonable distance, and subsequently optimized at the PBE0-D3/def2-TZVPP level of theory while freezing the rest of the atoms to refine the Cu-He bond distance. All calculations were performed using the Gaussian 16 program package.[46] The structural model for the Cu(I)/UiO-66 (Fig. 2g) was curated from reference,[42] which is based on the crystal structure of UiO-66.[41] As found in previous experimental reports, the Cu$^+$ ion is supported on a defective Zr$_6$ node of UiO-66 with a missing linker with the chemical formula of the node being Cu(I)/Zr$_6$O$_4$(OH)$_3$(linker)$_{11}$(OH)$_1$ where one Cu$^+$ ion and one hydroxyl group are placed to cap the defect site and a $\mu_3$ hydroxyl proton is removed to maintain the charge neutrality of the cluster. Here, the benzenedicarboxylate linkers were replaced by formates to reduce computational costs. The active site for He binding features a two-fold coordinated Cu(I), bound to an oxygen atom from the OH ligands capping the defect site and a nearby $\mu_3$ oxygen. The structural model is optimized at the PBE0-D3/def2SVP level of theory with the carbon atoms of the formate linkers fixed in order to mimic the constraints in the MOF environment. The vibrational mode decomposition analysis is performed using the open-source vibAnalysis package.[56]

The SBUs extracted from the HEALED library were capped with a protons at the node connection points. The proton positions were relaxed at PBE0-D3/def2SVP level of theory with the remainder of the cluster atoms fixed to their crystal positions. Two SBUs featuring bicoordinated Cu(I) sites, were referred as Cu$_3$Pyr$_3$ (pyr = pyrazolate) and Cu[(Cu$_4$Cl)(ttpm)$_2$]$_2$. Cu$_3$Pyr$_3$ (Fig. 2k), found in several copper pyrazolate-based MOFs, features a two-fold coordinated Cu(I) connected to pyrazolate ligands in a linear arrangement.[57] While the active Cu(I)-site in Cu[(Cu$_4$Cl)(ttpm)$_2$]$_2$ (Fig. 2h) found in WOLRIZ, adopts a bent geometry resulting from the coordination of two tetrazolate ligands.[45] On the other hand, for the defected paddlewheel SBUs, the corresponding SBU with the formula Cu2O8C4X4 (m2 and X being the node connection points) was extracted from the HEALED and capped with protons. One formate linker was removed to create the defected

Cu(I)Cu(II)(BTC)$_3$ (Fig. 2i) SBU. Furthermore, the one Cu$^{2+}$ is replaced by Zn$^{2+}$ to create the mixed metal Cu(I)Cu(II)(BTC)$_3$ (Fig. 2j) SBU. After all MOFs had been selected, a He atom was placed above the Cu(I) center and constrained geometry optimizations were carried out at the PBE0-D3/def2-SVP level to preserve the MOF environments. Furthermore, DLPNO-CCSD(T) single-point calculations were carried out in ORCA 5.0.1.[51] with the aug-cc-pVTZ basis set for Cu and He and the cc-pVTZ basis set for all other atoms. (see Fig. 2).

## Data availability

The molecular geometries and energies raw data generated in this study have been deposited in the zenodo database, under the accession code: https://zenodo.org/records/16281167. The processed data generated in this study is provided in the Supplementary Information and the Source Data file. Source data are provided with this paper.

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

## Acknowledgements

This work was funded by the Deutsche Forschungsgemeinschaft (DFG), Project ID 443871192 – GRK 2721: "Hydrogen Isotopes, $^{1,2,3}$H" (T.H.) and the European Research Council (ERC) Horizon grant agreement ID 101167472 "2D PolyMembrane" (T.H.). The authors thank the Center for Information Services and High Performance Computing (ZIH) at TU Dresden for computational resources. S.D. acknowledges funding from the Horizon Europe's Marie Skłodowska-Curie Actions (grant agreement ID. 101207236).

## Author contributions

Conceptualization T.H.; Methodology E.G.D., S.D., F.M., T.R.W., and T.H.; Investigation E.G.D, S.D., F.M.; Data Curation E.G.D, S.D., and F.M.; Analysis E.G.D, S.D., F.M., T.R.W., and T.H.; Writing- Original Draft, E.G.D., S.D., T.R.W., and T.H.; Writing-Review and Editing, E.G.D., S.D., F.M., T.R.W., and T.H.; Supervision, Project Administration, T.H.; Funding Acquisition, T.H. and S.D.

## Funding

## Competing interests

The authors declare no competing interests.
