## [Peer Review file · Nature Communications]

Prediction of strong Cu(I)–He interaction at open metal sites enables isotope-selective helium adsorption

Corresponding Author: Professor Thomas Heine

Version 0:

Reviewer comments:

Reviewer #1

(Remarks to the Author)

This study presents a theoretically driven investigation into the unconventional Cu(I)–He bonding and its potential for helium isotope separation. The authors employ high-level quantum-chemical calculations (e.g., CCSD(T), DLPNO-CCSD(T)) and QTAIM analysis to quantify the strength of polarization-driven Cu–He interactions and elucidate the underlying charge-transfer mechanisms. While the work highlights an innovative concept with potential applications in nuclear and cryogenic technologies, its scope is largely theoretical, relying on gas-phase clusters and static cluster models of MOFs rather than real-world material validations. The findings may appeal to niche audiences in computational chemistry and materials science but lack broader experimental or industrial relevance. As such, the paper may be more appropriate for a specialized journal focused on theoretical and computational studies rather than a multidisciplinary general journal.

Additional comments:

1. No experimental data (e.g., isotope-selective desorption experiments) is provided to corroborate the predicted ΔE (energy differences) or ΔZPE (zero-point energy) effects. Propose ion-trap experiments to directly measure Cu⁺–He complex stability and isotope-specific interactions. For materials-based studies, suggest temperature-programmed desorption (TPD) under isotopically enriched He mixtures (³He/⁴He 1–10%) to validate separation factors experimentally.
2. The study uses static cluster models (e.g., Cu[(Cu₄Cl)(tppm)₂]₂) to represent MOFs. However, real MOFs exhibit dynamic framework flexibility (e.g., vibrations, thermal expansion) that could alter local Cu–He distances and interactions. Future work should incorporate periodic DFT or molecular dynamics simulations to account for framework flexibility and its impact on adsorption behavior.
3. Perform path-integral molecular dynamics (PIMD) on periodic models of Cu[(Cu₄Cl)(tppm)₂]₂ and Cu(I)/UiO-66 at 20 K and 77 K to evaluate full quantum ZPE contributions (beyond harmonic approximations), He diffusion pathways and possible kinetic isotope effects (KIEs) in framework cages. Use grand-canonical Monte Carlo (GCMC) simulations (incorporating quantum corrections via the Feynman-Hibbs approach) to predict adsorption isotherms and loadings under realistic pressures. This would bridge quantum mechanical results with macroscopic separation performance.

Reviewer #2

(Remarks to the Author)

This manuscript presents a timely and well-executed computational study on isotope-selective helium adsorption at Cu(I) sites, identifying the Cu⁺–OH[−] coordination environment as particularly effective. The methodology is sound and the results are compelling, especially the predicted high separation factor for ³He/⁴He at cryogenic temperatures. The work offers valuable insight into the role of local electrostatics and provides a promising lead for future materials development.

However, the rationale for why Cu⁺–OH[−] outperforms other ligands remains largely qualitative. To support and generalize the observed trend, the authors are encouraged to include quantitative descriptors such as electrostatic potential maps or energy decomposition analyses to clarify how local electric field strength correlates with binding energy or ZPE splitting. Additionally, the ligand space explored is limited; expanding the comparison to include at least one representative from other anionic ligand families such as halides (F[−], Cl[−]), alkoxides (RO[−]), or thiolates (RS[−]) would enhance the design relevance of the conclusions. Since the analysis relies on 1D potential wells for nuclear quantum effects, a brief 3D treatment—such as a small-scale PIMD or quasi-harmonic estimate for the Cu⁺–OH[−] site—would help validate the magnitude of the predicted isotope selectivity. Finally, the discussion would benefit from articulating a more general design

principle, including possible thresholds for electric field strength or site accessibility, to guide future material development.

These are minor additions that could largely be addressed in supplementary material. With these clarifications, the study will offer both a solid mechanistic foundation and broadly useful insights for the design of He isotope separation adsorbents. I recommend minor revision.

Reviewer #3

(Remarks to the Author)

The paper reports on 3-H3/4-He gas separation on Cu(I) sites. What makes this paper publishable in Nature Commun. is the potential technological application. The methods used are state-of-the-art. In reading this paper I found however some few annoying problems which the authors need to address, otherwise the paper can be published.

Abstract: extremely high ionization potential, electron affinity and low polarizability. Extremely high electron affinity, really? The electron affinity of He is exactly zero. There may be a scattering state with a certain life time, but the electron will sit in the continuum.

Introduction: "and opens the door for alternative 3He production techniques", I guess separation is meant here and not production?

Again, "negligible electron affinity". It is exactly zero! He does not bind an electron!

Results and Discussion: An important paper has been overlooked, i.e. Frenking et al. J. Am. Chem. Soc. 1988, 110, 8007.

Here Frenking et al reports for the first time strong He interactions and conclude that the interaction is due to charge-induced dipole interactions. I would recommend to read this paper very carefully and make appropriate changes. So, the strong He-Cu(I) interactions are perhaps not so surprising at all and the usually strong polarization-driven interaction interaction is charge-induced dipole.

What I found somewhat annoying is the neglect of previous literature on He separation. For example, important papers they missed are 3-He/4-He separation by Schrier and Hauser J. Phys. Chem. Lett. 2010, 1, 2284–2287, J. Phys. Chem. Lett. 3, 209-213 (2012) and J. Phys. Chem. C 116, 10819-10827 (2012).

I believe He is inherently quantum and one may have to go beyond the Born-Oppenheimer approximation and include tunneling effects, but what is represented here is a good first start. Nevertheless it should be mentioned.

What is interesting is that The strongest interaction is found for the neutral Cu+(OH-) cluster which shows more than twice the adsorption energy compared to the bare Cu+ ion. I guess the charge-induced dipole interaction is very dependent on the charge and distance (R^{-4} behavior) to the Cu atom. Please check.

Version 1:

Reviewer comments:

Reviewer #1

(Remarks to the Author)

The authors have made careful revisions to the manuscript and addressed the concerns I raised; I have no further comments and recommend acceptance of this article for publication in Nature Communications.

Reviewer #2

(Remarks to the Author)

The authors have thoroughly addressed all of the reviewers' comments. I believe the manuscript is now ready for publication.

Reviewer #3

(Remarks to the Author)

The authors considered all the queries by the referees in an adequate manner. One item still needs attention. Reference (30) of Frenking et al has been inserted but no context given to the paper. Once this is corrected, the paper can be published.

Reviewer comments are in black, replies are in blue, and changes to the manuscript are in red.

Reviewer #1:

This study presents a theoretically driven investigation into the unconventional Cu(I)–He bonding and its potential for helium isotope separation. The authors employ high-level quantum-chemical calculations (e.g., CCSD(T), DLPNO-CCSD(T)) and QTAIM analysis to quantify the strength of polarization-driven Cu–He interactions and elucidate the underlying charge-transfer mechanisms. While the work highlights an innovative concept with potential applications in nuclear and cryogenic technologies, its scope is largely theoretical, relying on gas-phase clusters and static cluster models of MOFs rather than real-world material validations. The findings may appeal to niche audiences in computational chemistry and materials science but lack broader experimental or industrial relevance. As such, the paper may be more appropriate for a specialized journal focused on theoretical and computational studies rather than a multidisciplinary general journal.

We thank the reviewer for their assessment of the manuscript. We acknowledge that our approach employs gas-phase clusters and MOF cluster models; however, as shown in various other publications of our and other groups it is well-known that the long-range interaction of the periodic MOF lattice have only minor contributions to Langmuir-type adsorptions on strong sites (see below for details). Importantly, as noted below, we are currently working with Knut R. Asmis' group at the University of Leipzig to validate these theoretical predictions (see below for first results).

Additional comments:

1. No experimental data (e.g., isotope-selective desorption experiments) is provided to corroborate the predicted ΔE (energy differences) or ΔZPE (zero-point energy) effects. Propose ion-trap experiments to directly measure Cu⁺–He complex stability and isotope-specific interactions. For materials-based studies, suggest temperature-programmed desorption (TPD) under isotopically enriched He mixtures (³He/⁴He 1–10%) to validate separation factors experimentally.

We agree with the reviewer that ion-trap experiments can provide direct insight into $\text{Cu}^+\text{-He}$ stability and isotope effects. Such measurements were recently performed by the group of Prof. Knut R. Asmis at the University of Leipzig, and the results demonstrate that the $\text{Cu}^+\text{-He}$ interaction can be characterized. Moreover, we observe very good agreement between the experimental vibrational spectra of $\text{Cu}^+(\text{H}_2\text{O})\text{He}$ in the O–H stretching region ($3500\text{--}3800\text{ cm}^{-1}$) and the calculated frequencies when anharmonicity is included via VPT2 at the MP2/def2-TZVPP level. The planned publication, incorporating also other systems, is spectroscopically very demanding and rich in detail, so we abstain from including it in this work as it would defocus the discussion from the most important finding: the strong adsorption and selectivity between the helium isotopes. In addition, the interesting neutral species cannot be measured using this technique directly.

Concerning the MOF examples, it would indeed be nice to have the proposed measurements for the proposed materials, but this significant experimental effort is beyond this theoretical work.

Figure Caption: IRPD Spectra of $\text{Cu}^+(\text{H}_2\text{O})\text{He}$ (A), as well as the predicted infrared spectra (B) considering anharmonic contributions (VPT2/MP2/def2-TZVPP) as well as a 10 cm^{-1} wide Gaussian lineshape function. The rovibrational features (red dashed lines) are simulated using the PGOPHER program with rotational constants obtained from VPT2/MP2/def2-TZVPP calculations, the temperature is set at 100 K and linewidth of 8 cm^{-1} with Gaussian lineshape profile. Labeling of vibrational modes: ν_{OH}^s (symmetric stretching mode) and $\nu_{\text{OH}}^{\text{as}}$ (antisymmetric stretching mode),

2. The study uses static cluster models (e.g., Cu[(Cu₄Cl)(tppm)₂]₂) to represent MOFs. However, real MOFs exhibit dynamic framework flexibility (e.g., vibrations, thermal expansion) that could alter local Cu–He distances and interactions. Future work should incorporate periodic DFT or molecular dynamics simulations to account for framework flexibility and its impact on adsorption behavior.

In order to assess the effect of the framework vibrations on the Cu–He binding, we have computed the vibrational modes of the Cu/UiO-66 MOF cluster model at the PBE0-D3/def2SVP level of theory. From the computed vibrational spectrum, we have decomposed the atomic motions in terms of the displacements along a set of internal coordinates that reveal the most prominent ones involved in the framework vibrations. This vibrational mode decomposition analysis reveals that the Cu–He stretch is highly localized, with negligible coupling to the framework vibrations. The Cu–He bond contributes significantly to only one low-frequency mode (132.3 cm⁻¹), while all other modes display minimal Cu–He involvement (<3%). This indicates that the framework motions are dynamically independent of the Cu–He vibration, and therefore, the Cu–He bond length and interaction are not significantly affected by other molecular vibrations. On the other hand, previous ab initio and IR analysis for UiO-66 suggest that the framework phonons and skeletal vibrations are mainly localized in the 400-800 cm⁻¹ region (μ₃O, Zr-O, linker bending stretching). This energetic separation indicates minimal coupling between the lattice modes and the Cu-He vibration. Thus, these results collectively suggest that the Cu–He bond length/interaction is not significantly modulated by lattice or framework vibrations.

We have added the following text on page 12 of the revised manuscript, and vibrational mode decomposition analysis is included in the Supplementary Text 4 and Fig. S7.

Furthermore, vibrational mode decomposition analysis confirms that the Cu-He stretch is highly localized, with negligible coupling to the node or lattice vibrations, indicating that the Cu-He interaction is not expected to be significantly modulated by framework dynamics. (see Fig. S7). Note that the same conclusions have been drawn earlier for the adsorption of stronger interacting dihydrogen in various MOFs, e.g. in Cu(I)-MFU-4l.(26)

On page 18 in the Methods section we added the following text:

The vibrational mode decomposition analysis is performed using the vibAnalysis package.

Fig. S7. Decomposition of the vibrational mode at 132.3 cm^{-1} for the Cu(I)/UiO-66 cluster (only top ten contributions shown), highlighting the Cu–He stretch as the major contributing motion.

We would like to point out that the same conclusions have been drawn earlier for the adsorption of stronger interacting dihydrogen in various MOFs, e.g. in Cu(I)-MFU-4l.

3. Perform path-integral molecular dynamics (PIMD) on periodic models of $\text{Cu}[(\text{Cu}_4\text{Cl})(\text{ttpm})_2]_2$ and Cu(I)/UiO-66 at 20 K and 77 K to evaluate full quantum ZPE contributions (beyond harmonic approximations), He diffusion pathways and possible kinetic isotope effects (KIEs) in framework cages. Use grand-canonical Monte Carlo (GCMC) simulations (incorporating quantum corrections via the Feynman-Hibbs approach) to predict adsorption isotherms and loadings under realistic pressures. This would bridge quantum mechanical results with macroscopic separation performance.

Our quantum chemical calculations inherently account for anharmonic effects through a numerical treatment that includes the one-dimensional Cu–He stretch coordinate along with the perpendicular vibrational motions. The reliability of this approach has been further validated for representative Cu(I) model complexes by comparison with both VPT2 and DVR calculations (for further details, see Methods section in the main text and Supplementary Texts S2 and S5 in the Supplementary Materials), as

discussed in the revised manuscript. The corresponding quantum zero-point energy (ZPE) estimates obtained from this numerical scheme for all material models are summarized in Table S10 in the Supplementary Materials.

PIMD and GCMC calculations based on classical or semiclassical assumptions such as the force fields themselves and the Feynman-Hibbs approximation are inferior in accuracy.

For dynamic studies, PIMD would be a suitable method, in particular when going beyond the Langmuir model. However, this requires an accurate force field to be performed in a computationally tractable fashion. Note that one of the important results from our paper is that there is a change in the nature of the Cu-He bond when we have different ligands. For instance, the fact that the Cu-He bond is stronger in Cu+(F-)He than in Cu+He would not be captured by a standard force field and additional parameterization or machine learned potentials would have to be developed, which is beyond the scope of this work.

Given the small kinetic diameter of He (2.6 Å) and its very weak, non-specific interactions with the framework, we expect only a negligible diffusion barrier for He in MOFs—particularly in systems such as Cu[(Cu₄Cl)(tppm)₂]₂ and Cu(I)/UiO-66, where the pore apertures (e.g. it is 6.0 Å for Cu(I)/UiO-66 and even larger in Cu[(Cu₄Cl)(tppm)₂]₂) are significantly larger than the kinetic diameter of helium.

We have added the following lines in the revised manuscript on page 15 regarding the possible future work suggested by the referee

To go beyond the Langmuir model employed in this work, further studies require multi-particle effects, including He tunneling between metal sites and fermion (³He)/boson (⁴He) spin effects on adsorption.

Reviewer #2:

This manuscript presents a timely and well-executed computational study on isotope-selective helium adsorption at Cu(I) sites, identifying the Cu⁺–OH[−] coordination environment as particularly effective. The methodology is sound and the results are compelling, especially the predicted high separation factor for ³He/⁴He at cryogenic

temperatures. The work offers valuable insight into the role of local electrostatics and provides a promising lead for future materials development.

We thank the reviewer for their review of the manuscript and encouraging comments.

1. However, the rationale for why $\text{Cu}^+\text{-OH}^-$ outperforms other ligands remains largely qualitative. To support and generalize the observed trend, the authors are encouraged to include quantitative descriptors such as electrostatic potential maps or energy decomposition analyses to clarify how local electric field strength correlates with binding energy or ZPE splitting. Additionally, the ligand space explored is limited; expanding the comparison to include at least one representative from other anionic ligand families such as halides (F^- , Cl^-), alkoxides (RO^-), or thiolates (RS^-) would enhance the design relevance of the conclusions.

We followed the reviewer's recommendation and, in the revised manuscript, performed additional computations on four more model complexes to gain deeper insights into Cu-He binding trends. In total, eight model Cu(I) complexes were analyzed for their He-binding strengths and $^4\text{He}/^3\text{He}$ separation selectivities. We further carried out energy decomposition analysis (EDA-NOCV) for all model systems to better understand the underlying factors governing He binding and to correlate various local electronic descriptors with the binding energies. A subset of these new results is discussed in the main text of the revised manuscript and presented in Table 1 and Fig. 1, while the complete dataset—including all EDA-NOCV results—is provided in Tables S2 in the Supporting Information. Among the model complexes, $\text{Cu}^+(\text{F}^-)$ exhibits the strongest He binding with a binding energy of -19.4 kJ/mol. The EDA-NOCV analysis shows that, within the neutral complexes bearing anionic ligands, the binding energy follows the electronegativity order of the donor atom bonded to Cu: lower electronegativity leads to weaker binding, consistent with the decreasing electrostatic contribution to the total interaction energy. Accordingly, Figures 1 and 3 have been updated to include the results for $\text{Cu}^+(\text{F}^-)$.

We appreciate the reviewer's recommendation to explore additional systems. Our analysis revealed that $\text{Cu}^+(\text{F}^-)$ delivers the highest separation factor, and this result has been integrated into the updated manuscript.

In the revised manuscript, the text in Pg 4, Fig. 1A, Table 1 and Fig. 3 are accordingly modified after including the result of $\text{Cu}^+(\text{F}^-)$. The revised text in Pg 4 now reads as:

We have selected eight $\text{Cu}(\text{I})$ complexes for initial conceptual and benchmark studies: two charged clusters Cu^+He and $\text{Cu}^+(\text{H}_2\text{O})\text{He}$, which would be amenable to assessment in ion trap experiments as demonstrated for dihydrogen isotopes earlier, (28, 29) six uncharged ones, $\text{Cu}^+\text{X-He}$ ($\text{X} = \text{F}, \text{Cl}, \text{Br}, \text{OH} \ \& \ \text{SH}$), and $\text{Cu}^+(\text{H}_2\text{O})(\text{OH}^-)\text{He}$ (Fig. 1A). All structures have been fully optimized at the CCSD(T)/aug-cc-pVTZ level of theory. In all eight model complexes we observe a strong Cu-He interaction. This is reflected in Born-Oppenheimer (BO) Cu-He bond distances ranging from 1.70 to 2.00 Å and BO adsorption energies (BSSE corrected) of -7.6 to $-19.4 \text{ kJ mol}^{-1}$. The strongest interaction is found for the neutral $\text{Cu}^+(\text{F}^-)$ complex, which shows more than twice the adsorption energy compared to the bare Cu^+ ion (Fig. 1A, Table 1).

Here is the revised Fig. 1

Fig. 1. Accurate quantum chemical calculations of Cu-He interactions in $\text{Cu}(\text{I})$ complexes. A) Structures of the Cu^+He , $\text{Cu}^+(\text{F}^-)\text{He}$, $\text{Cu}^+(\text{OH}^-)\text{He}$, $\text{Cu}^+(\text{H}_2\text{O})\text{He}$ and $\text{Cu}^+(\text{H}_2\text{O})(\text{OH}^-)\text{He}$ complexes and results of the QTAIM analysis. Cu-He bond lengths (CCSD(T)/aug-cc-pVTZ level) are shown next to the bond paths and critical points as obtained by the QTAIM analysis, while Hirshfeld charges are shown for Cu (in brown) and He (in light blue) (PBE0-D3(BJ)/TZ2P level on CCSD(T)/aug-cc-pVTZ geometries). Colour scheme: Cu—brown, O—red, He—light blue, H—white. Bond critical points are shown as small green spheres. Values of the electron density ρ_{BCP} and its Laplacian $\nabla^2\rho_{\text{BCP}}$ are reported in atomic units (a.u.). **B)** Plot of the potential energy along the Cu-He stretch coordinate. All values are at

the BSSE-corrected CCSD(T)/aug-cc-pVTZ level. ZPE-corrected adsorption energies (numerical method) are given for ^3He (dotted lines) and ^4He (dashed lines). **C**) DVR ^4He (light blue mesh) and ^3He (orange mesh) nuclear density contours of 99% cumulative probability, for Cu^+He , $\text{Cu}^+(\text{F})\text{He}$ and $\text{Cu}^+(\text{H}_2\text{O})\text{He}$. **D**) Predicted separation factors for adsorptive He isotopes at low temperatures. All values are based on the CCSD(T)/aug-cc-pVTZ level (solid line) and DLPNO-CCSD(T)/cc-pVTZ(cu-He: aug-cc-pVTZ) (dashed line).

Here is the revised Table 1:

Table 1. Cu–He interaction analysis. Adsorption energies E_{ads} , and Cu-He bond lengths (d_{CuHe}) at the BSSE-corrected CCSD(T)/aug-cc-pVTZ and at the DLPNO-CCSD(T)/cc-pVTZ(aug-cc-pVTZ for Cu and He) levels of theory (values given in the **square brackets**). Nuclear quantum effects are accounted for using the numerical approach (see text). For comparison, ZPE corrections using the harmonic approximation are given for the CCSD(T) calculations. Hirshfeld charges for Cu in the systems with adsorbed helium q_{CuHe} and for the adsorbed He atom q_{He} , electron density ρ_{BCP} at the BCP between Cu^+ and He. The ZPE difference is also benchmarked for calculating $\text{Cu}^+\text{-He}$ adsorption energies. All distances are expressed in Å and energy values in $\text{kJ}\cdot\text{mol}^{-1}$.

Systems	$E_{\text{ads}}^{\text{BO}}$	d_{CuHe}		E_{ads}^0 (analytical)		E_{ads}^0 (numerical)		q_{CuHe} (e)	q_{He} (e)	ρ_{BCP} ($e \cdot r_{\text{Bohr}}$)	ΔZPE	
		^3He	^4He	^3He	^4He	^3He	^4He				analytical	numerical
Cu^+He	-7.6 [-7.2]	2.08 [2.07]	2.07 [2.06]	-6.0 [-]	-6.2 [-]	-6.1 [-5.7]	-6.3 [-5.9]	+0.900 [+0.897]	+0.101 [+0.103]	0.032 [0.034]	0.20 [-]	0.18 [0.19]
$\text{Cu}^+(\text{F})\text{He}$	-19.4 [-18.6]	1.74 [1.73]	1.73 [1.72]	-13.8 [-]	-14.6 [-]	-14.1 [-13.1]	-14.8 [-13.8]	+0.319 [+0.332]	+0.112 [+0.096]	0.067 [0.036]	0.74 [-]	0.68 [0.68]
$\text{Cu}^+(\text{OH}^+)\text{He}$	-17.1 [-16.0]	1.77 [1.76]	1.76 [1.75]	-11.7 [-]	-12.4 [-]	-11.9 [-10.5]	-12.6 [-11.2]	+0.265 [+0.264]	+0.106 [+0.108]	0.062 [0.066]	0.71 [-]	0.64 [0.67]
$\text{Cu}^+(\text{H}_2\text{O})\text{He}$	-14.4 [-13.1]	1.86 [1.85]	1.85 [1.84]	-10.7 [-]	-11.2 [-]	-10.5 [-9.1]	-11.0 [-9.6]	+0.599 [+0.597]	+0.120 [+0.122]	0.050 [0.052]	0.49 [-]	0.48 [0.49]
$\text{Cu}^+(\text{H}_2\text{O})$ (OH $^-$)He	-12.1 [-11.0]	1.82 [1.80]	1.81 [1.79]	-7.2 [-]	-8.2 [-]	-7.2 [-6.2]	-7.7 [-6.8]	+0.284 [+0.278]	+0.100 [+0.104]	0.055 [0.059]	0.59 [-]	0.61 [0.62]

Here is the revised Fig. 3:

Fig. 3: Separation factors of top-performing Cu(I)-hosting materials for He isotope separation. Predicted separation factors for adsorptive He isotope on selected Cu(I)-candidates, plotted as a function of temperature.

Furthermore, In the revised manuscript on Pg. 5 we added the following text and Table S2

In addition, energy decomposition analyses (EDA-NOCV)(32,33) reveal that, within the neutral complexes bearing anionic ligands, the binding energy follows the electronegativity order of the donor atom bonded to Cu: lower electronegativity leads to weaker binding, consistent with the decreasing electrostatic contribution to the total interaction energy (see Table S2).

In the revised manuscript on Pg 7, we modified the text to include the $\text{Cu}^+(\text{F}^-)$ results

The pronounced ΔZPE result in $^4\text{He}/^3\text{He}$ separation factors reaching 39.5 at liquid hydrogen (LH2) and even 2.0 at liquid nitrogen (LN2) temperatures for $\text{Cu}^+(\text{F}^-)\text{He}$ (Fig. 1C and Table S5).

Finally in the abstract, on Pg 1, we modified the text as following

We first accurately calculate the interaction of Cu(I) with helium in eight gas phase clusters and elucidate the nature of the Cu-He interaction

2. Since the analysis relies on 1D potential wells for nuclear quantum effects, a brief 3D treatment—such as a small-scale PIMD or quasi-harmonic estimate for the $\text{Cu}^+ - \text{OH}^-$ site—would help validate the magnitude of the predicted isotope selectivity.

First, we would like to clarify that in our submitted version, the NQEs were described with a 3D treatment, a numerical FDM for the Cu-He stretch, and a harmonic oscillator term for the He out-of-plane modes. In the revised version, we have modified our approach to include anharmonicity corrections also in the out-of-plane modes, via perturbation theory and a quartic potential fit. We also added in the supporting information discrete-variable-representation (DVR) calculations- which are highly accurate solutions to the 3D Schrödinger equation for He in the BO potential of the Cu complexes- to validate the separation between the stretch and out-of-plane modes, and the computation of the partition function.

We modified the text on Pg 5-6 as follows,

To analyze the NQEs on He adsorption, vibrational frequencies have first been assessed analytically using the harmonic approximation (Table S3). However, the potential energy surface (PES) of the Cu-He coordinate has a strongly anharmonic character (Fig. 1B), leading to asymmetric distributions of the He nuclear densities along the Cu-He axis (Fig. 1C). Moreover, the Cu-He bond dissociation curve expectedly shows the strongly anharmonic character of weak bonds, with the different asymptotic character of the charge neutral (London dispersion governed $\sim r^{-6}$ asymptotics) compared to the charged (charge-induced dipole governed $\sim r^{-4}$ asymptotics) clusters. As standard methods to account for anharmonicity are computationally unfeasible for the realistic material models, we developed an approach focused on the most significant degrees of freedom for He adsorption: the Cu-He bond stretch and the two He displacements normal to this bond. For the Cu-He stretch NQEs, first we fitted the 1D PES along this coordinate to a Morse potential augmented with a charge-induced dipole term, and then solve the 1D Schrödinger equation numerically using the finite difference method (FDM) (36, 37). The NQEs of the two He degrees of freedom normal to the Cu-He bond were described using a quartic anharmonic correction to a harmonic oscillator potential. We call the combination of the FDM plus quartic NQE models the “*numerical*” approach in the remainder of the document. The results obtained with this numerical approach have been validated

against second order vibrational perturbation theory (VPT2) and discrete variable representation (DVR) calculations (for details see Supplementary Texts 2 and 5)

To accommodate the DVR calculations in the revised manuscript, we have rewritten the methods section on **Nuclear quantum effects and isotope separation factors**. On Pg 16-17 we modified the text as below. Accordingly, Fig. 1 (already shown above) and Table 3 is also modified to include the DVR results.

Nuclear quantum effects and isotope separation factors

For the NQE calculations accounting for anharmonicity, we only consider three degrees of freedom (DOF), namely those related to the helium motion. For the stretch vibration, we first fitted the PES of the Cu-He bond, from 1.4 Å to bond dissociation, to a one-dimensional Morse potential with a charge-induced dipole (CID) term, which is switched off for short distances via a Fermi function (see details in Supplementary Text 1 and for parameters and shape, see Fig. S1, Table S14, S16 and S18). The finite difference method (FDM) (36, 37) was then applied to solve the one-dimensional Schrödinger equation using the reduced mass of the Cu-He moiety (see details in the Supplementary Text 2). The increase of Cu-He bond length is an additional result of the FDM calculation. For the two He normal modes to the Cu-He bond, the potential was fitted to a quartic polynomial, via a five-point fit. From this fit, a quartic anharmonicity correction term to the harmonic ZPE was derived (see details in Supplementary Text 2, and for parameters, see Table S15, S17 and S19).

This protocol to calculate the numerical frequencies has been validated with second-order vibrational perturbation theory (VPT2) and discrete variable representation (DVR) calculations. The VPT2 analysis was performed at the MP2/def2-TZVPP level to assess the magnitude of the anharmonicity in the vibrational frequencies and the impact of considering only the three DOF related to the helium motion. The MP2 calculations have been performed using Gaussian 16 (46). The DVR calculations, which numerically solve the 3D Schrodinger equation, were performed with grids of CCSD(T)/aug-cc-pVTZ energies. The DVR results were employed to validate the separation of the Cu-He stretch and the two He normal modes. The NuSol package (53) was employed for the DVR calculations. Table 3 presents a comparison of the vibrational frequencies for four systems. The complete validation analysis is found in the Supplementary Text 5.

The $^4\text{He}/^3\text{He}$ separation factors ($^4\text{He}/^3\text{He}$) are calculated via the equilibrium adsorption constants (^3He) and (^4He). These adsorption constants are obtained from the partition functions which have been calculated by multiplying the individual contributions from each of the three normal modes based on the Langmuir model and the ideal gas approximation for translational partition function of free He (see details in the Supplementary Text 3),

$$\alpha(^4\text{He}/^3\text{He}) = \frac{K_{ad}(^4\text{He})}{K_{ad}(^3\text{He})} \quad \text{where. } K_{ad}(^4\text{He}) = \frac{q_{^4\text{He}}^{ad}}{q_{^4\text{He}}^g} \quad \text{and } K_{ad}(^3\text{He}) = \frac{q_{^3\text{He}}^{ad}}{q_{^3\text{He}}^g} . . .$$

Here is the revised Table 3

Table 3. Calculated vibrational frequencies in Cu(I) complexes. Harmonic, anharmonic VPT2 and FDM plus quartic numerical frequencies (cm^{-1}) of ^4He in small Cu(I)-complexes computed at MP2/def2-TZVPP level of theory. FDM plus quartic numerical and 3D-DVR frequencies at the CCSD(T)/ aug-cc-pVTZ level of theory are also given^a.

Level	MP2 def2-TZVPP						CCSD(T) aug-cc-pVTZ	
Systems	Modes	ν_{harm}^{full}	ν_{harm}^{He}	ν_{VPT2}^{full}	ν_{VPT2}^{He}	ν_{FDMq}^{He}	ν_{FDMq}^{He}	$\nu_{DVR}^{He, b}$
Cu ⁺ He	ν_s	187.4	187.4	134.1	134.1	140.1	110.7	111.0
Cu ⁺ (F ⁻)He	ν_s	518.9	527.8	427.9	431.9	495.6	430.9	401.1
	$\nu_{\perp 1}$	215.7	180.1	199.1	168.8	185.8	159.0	150.5
	$\nu_{\perp 2}$	215.7	180.1	199.1	168.8	185.8	159.0	150.5
Cu ⁺ (H ₂ O)He	ν_s	337.9	359.4	272.3	277.2	301.2	324.8	316.9
	$\nu_{\perp 1}$	118.7	101.0	109.0	89.5	100.4	100.7	84.7
	$\nu_{\perp 2}$	122.4	103.8	138.3	92.5	107.0	101.2	87.9

^a ν^{full} : frequencies obtained with full normal modes. ν^{He} : keeps only the three degrees of freedom of the adsorbed He atom. ν_{FDMq}^{He} : are the frequencies obtained using the FDM plus quartic numerical approach. ν_s represents the Cu-He stretching vibrational frequency, and $\nu_{\perp 1}$ and $\nu_{\perp 2}$ correspond to the orthogonal contribution modes of He atom

^b3D-DVR frequencies are obtained as energy differences between the He vibrational ground state and excited states displaying a nodal plane perpendicular (ν_s) or parallel (ν_{\perp}) to the Cu-X axis.

According, the Table 2 is also slightly modified to include the results from the anharmonicity corrections in the out-of-plane modes

Table 2. Geometrical parameters of He interacting with models of host materials. Bond lengths and energies at the DLPNO-CCSD(T)/cc-pVTZ (aug-cc-pVTZ for Cu, He) level of theory. For Cu(I)/UiO-66-He an effective core potential (ECP) was used for Zr atom. QTAIM properties calculated at the PBE0-D3(BJ)/TZ2P level; Hirshfeld Charges of the cluster models are given with adsorbed He q_{CuHe} (the charges of adsorbed He atom q_{He} are given in the square brackets). Electron Density ρ_{BCP} and its Laplacian ∇^2_{BCP} at the BCP between Cu^+ and He.

Systems	d_{CuHe} (Å)		E_{ads}^0 (kJ·mol ⁻¹)		q_{CuHe} (e)	ρ_{BCP}	∇^2_{BCP}
	³ He	⁴ He	³ He	⁴ He	[q_{He} (e)]	($e \cdot r_{\text{Bohr}}$)	($e \cdot r_{\text{Bohr}}$)
6c2	1.90	1.89	-9.1	-9.5	+0.481 [+0.110]	0.052	0.334
9c3	1.97	1.96	-5.7	-6.0	+0.395 [+0.093]	0.042	0.261
2z'	2.23	2.21	-2.4	-2.5	+0.417 [+0.067]	0.023	0.130
3z'	2.20	2.18	-3.1	-3.3	+0.458 [+0.072]	0.025	0.146
Cu(I)-MFU-4l	2.94	2.92	-1.0	-1.1	+0.267 [+0.019]	0.006	0.026
Cu(I)Porphyrin	3.17	3.15	-2.1	-2.1	+0.583 [+0.023]	0.003	0.012
Cu(I)Cu(II)(BTC)₃	2.76	2.74	-1.3	-1.5	+0.451 [+0.034]	0.007	0.030
Cu(I)Zn(II)(BTC)₃	3.12	3.10	-0.4	-0.5	+0.327 [+0.022]	0.004	0.014
Cu(I)/UiO-66	2.35	2.33	-1.9	-2.1	+0.354 [+0.053]	0.014	0.071
Cu[(Cu₄Cl)(tppm)₂]	2.11	2.09	-3.9	-4.1	+0.406 [+0.085]	0.032	0.186

3. Finally, the discussion would benefit from articulating a more general design principle, including possible thresholds for electric field strength or site accessibility, to guide future material development.

In the revised manuscript, we modified the text on Pg 14 to articulate a more general design principle as derived from our study

However, this value is three times lower than that reported for small gas-phase complexes containing Cu(I) — with a record high of $-19.4 \text{ kJ mol}^{-1}$ for $\text{Cu}^+(\text{F}^-)$. This comparison indicates that further exploration may uncover MOFs featuring active, undercoordinated metal sites that are even more promising for adsorptive He isotope separation. Notably, our results highlight that two-coordinated open metal sites in bent geometries show the highest potentials among the MOF models investigated in this work.

Reviewer #3 :

The paper reports on 3-H3/4-He gas separation on Cu(I) sites. What makes this paper publishable in Nature Commun. is the potential technological application. The methods used are state-of-the-art. In reading this paper I found however some few annoying problems which the authors need to address, otherwise the paper can be published.

We thank the reviewer for their review and positive evaluation of the manuscript.

1. Abstract: extremely high ionization potential, electron affinity and low polarizability. Extremely high electron affinity, really? The electron affinity of He is exactly zero. There may be a scattering state with a certain life time, but the electron will sit in the continuum.

In the revised version of the manuscript, we have corrected this and modified the following sentence in the abstract

Helium is known as an inert element due to its extremely high ionization potential, zero electron affinity, and low polarizability.

2. Introduction: “and opens the door for alternative 3He production techniques”, I guess separation is meant here and not production?

We have corrected this sentence in the revised manuscript. The corresponding sentence on Pg 3 now reads as

This discovery is of fundamental interest to chemistry and opens the door for alternative ^3He separation techniques.

3. Again, “negligible electron affinity”. It is exactly zero! He does not bind an electron!

We have corrected this sentence in the revised manuscript. The corresponding sentence on Pg 3 now reads as

Despite mechanical trapping (caging),¹ chemical interactions of helium seem to be impossible due to its low polarizability of $0.204 \times 10^{-24} \text{ cm}^3$, about 4 times smaller than that of H_2 , its enormously high ionization potential of almost 25 eV, and its zero electron affinity.

4. Results and Discussion: An important paper has been overlooked, i.e. Frenking et al. J. Am. Chem. Soc. 1988, 110, 8007. Here Frenking et al reports for the first time strong He interactions and conclude that the interaction is due to charge-induced dipole interactions. I would recommend to read this paper very carefully and make appropriate changes. So, the strong He-Cu(I) interactions are perhaps not so surprising at all and the usually strong polarization-driven interaction interaction is charge-induced dipole.

We thank the reviewer for drawing our attention to the work by Frenking *et al.* In the revised manuscript, we have cited this work as reference (30). Furthermore, we have modified the text on Pg 5, which now reads as

The strong Cu–He bonding interactions are revealed by a quantum theory of atoms in molecules (QTAIM) analysis (PBE0-D3(BJ)/TZ2P, see Methods and Table S1), the bond critical points are located halfway between Cu and He.(30)

5. What I found somewhat annoying is the neglect of previous literature on He separation. For example, important papers they missed are 3-He/4-He separation by Schrier and Hauser *J. Phys. Chem. Lett.* 2010, 1, 2284–2287, .*J. Phys. Chem. Lett.* 3, 209-213 (2012) and *J. Phys. Chem. C* 116, 10819-10827 (2012).

We fully agree with the reviewer that these are relevant literature on He separation and apologize for not citing them earlier. We added the following text in the revised version of the manuscript and included these references (15-17) on Pg 3

Prior theoretical studies reported the possibility of ³He separation from ⁴He by exploiting kinetic quantum sieving at low temperature ($\leq 20\text{K}$) using nanoporous membranes, specifically nitrogen-functionalized graphene pores.(15-17)

6. I believe He is inherently quantum and one may have to go beyond the Born-Oppenheimer approximation and include tunneling effects, but what is represented here is a good first start. Nevertheless it should be mentioned.

We agree with the reviewer that, as a consequence of their light mass, the isotopes of He display significant quantum effects. With the protocol employed in the manuscript, we capture the zero-point energy and anharmonicity that arises from the NQEs. We have further validated our results by adding discrete variable representation calculations, which are highly accurate solutions to the 3D Schrödinger equation for He in the BO potential of the Cu complexes. Accordingly, we have modified Fig. 1 in the revised manuscript to add a panel depicting the probability density distribution of the quantum He atoms (Fig. 1c)

However, given the magnitude of the adsorption energies we found, we do not expect to observe significant non-BO effects. For instance, in the HeH⁺ molecular cation, the dissociation energy is decreased by 7.6 cm⁻¹ (0.01 kJ/mol) by the inclusion of adiabatic correction to account for non-BO effects [Bishop, David M., and Lap M. Cheung. "A theoretical investigation of HeH⁺." *Journal of Molecular Spectroscopy* 75.3 (1979): 462-473]. This corresponds to 0.5% of the HeH⁺ ground state vibrational energy. A number of this magnitude is too small to have an impact on the adsorption energies we have computed for the Cu-He complexes.

Similarly, we do not expect that considering tunneling would change the predicted separation factors. It is possible that tunneling corrections would modify the He adsorption/desorption rates, such that the system reaches the equilibrium condition faster. However, the separation factors were computed from equilibrium adsorption constants, which are not affected by adsorption kinetics, as detailed in the Supplementary Text 3. In future studies where we simulate the diffusion of He between different Cu sites, the tunneling effects will be relevant. Furthermore, in simulations with several He atoms, there might be significant quantum effects that arise from the spin statistics of ^3He (a fermion) and ^4He (a boson). We have added a future work paragraph on Pg 15, which includes the following lines regarding the quantum nature of He:

To go beyond the Langmuir model employed in this work, further studies require multi-particle effects, including He tunneling between metal sites and fermion (^3He)/boson (^4He) spin effects on adsorption.

The revised Figure 1 is shown above in the answers to Referee 1.

7. What is interesting is that The strongest interaction is found for the neutral $\text{Cu}^+(\text{OH}^-)$ cluster which shows more than twice the adsorption energy compared to the bare Cu^+ ion. I guess the charge-induced dipole interaction is very dependent on the charge and distance (R^{-4} behavior) to the Cu atom. Please check.

We appreciate the reviewer's remark highlighting the role of the charge-induced dipole interaction. In the revised version of the manuscript, we added the following sentence.

The Cu-He bond dissociation curve expectedly shows the strongly anharmonic character of weak bonds, with the different asymptotic character of the charge neutral (London dispersion governed $\sim r^{-6}$ asymptotics) compared to the charged (charge-induced dipole governed $\sim r^{-4}$ asymptotics) clusters.

Additional changes

1. During this revision, we have corrected a few grammatical issues, typos in the manuscript.
2. The Acknowledgements and Author Contributions sections are also slightly modified to maintain consistency.